# Role of Wnt Signaling in Mouse Fetal Skin Wound Healing

**DOI:** 10.3390/biomedicines10071536

**Published:** 2022-06-28

**Authors:** Kento Takaya, Ayano Sunohara, Noriko Aramaki-Hattori, Shigeki Sakai, Keisuke Okabe, Hideko Kanazawa, Toru Asou, Kazuo Kishi

**Affiliations:** 1Department of Plastic and Reconstructive Surgery, Keio University School of Medicine, 35 Shinanomachi, Shinjuku-ku, Tokyo 160-8582, Japan; a.sunohara@gmail.com (A.S.); nonken@2001.jukuin.keio.ac.jp (N.A.-H.); shigekix724@hotmail.com (S.S.); dawndawn@hotmail.co.jp (K.O.); mori@ideajapan.com (T.A.); kkishi@a7.keio.jp (K.K.); 2Faculty of Pharmacy, Keio University, 1-5-30 Shibakoen, Minato-ku, Tokyo 105-8512, Japan; kanazawa-hd@pha.keio.ac.jp

**Keywords:** regeneration, Wnt signalling, scar formation

## Abstract

Wnt proteins secrete glycoproteins that are involved in various cellular processes to maintain homeostasis during development and adulthood. However, the expression and role of Wnt in wound healing have not been fully documented. Our previous studies have shown that, in an early-stage mouse fetus, no scarring occurred after cutaneous wounding, and complete regeneration was achieved. In this study, the expression and localization of Wnt proteins in a mouse fetal-wound-healing model and their associations with scar formation were analyzed. Wnt-related molecules were detected by in-situ hybridization, immunostaining, and real-time polymerase chain reaction. The results showed altered expression of Wnt-related molecules during the wound-healing process. Moreover, scar formation was suppressed by Wnt inhibitors, suggesting that Wnt signaling may be involved in wound healing and scar formation. Thus, regulation of Wnt signaling may be a possible mechanism to control scar formation.

## 1. Introduction

Currently, there is no mechanism to regenerate wounded skin without scar formation in adult animals. When a wound is made on the skin, scar formation always occurs. In general, adult animals can only regenerate a limited number of organs, such as the liver. However, even in mammals, fetuses have been reported to possess the ability to regenerate skin until a certain developmental stage [1].

In this study, we focused on skin regeneration in mouse fetuses. During early development, even if the fetus is wounded, the skin is completely regenerated, including the arrangement of the dermal extracellular matrix (ECM), regeneration of hair follicles, and skin texture. However, during the late developmental stages, the skin does not regenerate and forms scars, exhibiting the same wound-healing phenotype as adult animals. Our studies, to date, have shown that this change in the way healing occurs is not gradual and that there is a tipping point [2]. We observed that the entire structure can regenerate up to embryonic day (E) 13, including the texture of the skin surface, but after E14, the texture of the skin disappears, leaving a visible mark. Further, the dermal structure regenerates until E16, but after E17, the dermis becomes fibrotic and leaves a scar. To further understand the mechanism of skin regeneration, it is important to focus on the expressions of genes that are altered before and after these transition points.

Regeneration may involve factors related to development and epidermal dermal interaction. During morphogenesis, intercellular communication occurs through interaction via secretory signal molecules, including the Wnt signaling pathway. Wnt genes have been shown to exhibit both temporal and positional expression during development and are known as inducers of morphogenesis, determinants of cell polarity, and regulators of proliferation and differentiation [3,4,5]. Wnt is also an important factor in skin development and is involved in multiple processes, including dermal development and formation of skin appendages, such as hair [6]. Canonical Wnt/β-catenin signaling is required for the identification of the mouse dermis [7,8]. In mouse ventral dermatogenesis, β-catenin is also involved in the survival of early ventral dermal progenitor cells. β-catenin protein levels and transcriptional activity are elevated in dermal fibroblasts during the proliferative phase of mouse skin wound healing and return to baseline during the remodeling phase [9].

Wnt and/or β-catenin signaling is believed to play important roles in various aspects of cutaneous wound repair and functions in skin development, including involvement in the construction of epithelial structures and reconstruction of dermal compartments. Large wounds in mice regenerate new hair follicles (HF) at the center. This phenomenon of wound-induced hair follicle neogenesis (WIHN) recapitulates the embryonic HF morphogenesis program and involves standard Wnt signaling [10,11,12]. In particular, both dermal and epidermal Wnt ligands are important and may act at different stages of WIHN [10,13]. Furthermore, forced expression of Wnt or Wnt signaling molecules stimulates cell proliferation, migration, and ECM degradation, reflecting the stages of wound healing [14]. Thus, the role of Wnt/β-catenin signaling in wound epithelial remodeling is being slowly elucidated, but its investigation at the embryonic developmental stage and its involvement in complete skin regeneration, including texture, is still unclear.

In this study, we investigated the behavior of Wnt/β-catenin in the wound using mouse fetal-wound-healing models. We examined the β-catenin pathway at 24 h post-wounding, when epithelialization has not yet been completed, to determine whether there are changes in expression during the transition from regeneration to repair. Specifically, the β-catenin-dependent Wnt family Axin2, involved in phosphorylation and degradation of β-catenin [14], and lymphoid-enhancer-binding factor 1 (LEF1) that forms a complex with β-catenin and promotes transcriptional activity [15], were examined. Further, we evaluated changes that occurred during skin regeneration by altering the Wnt signaling pathway. Three pharmacological agents were used to regulate Wnt signaling, 6-Bromoindirubin-3′-oxime (6-BIO) as an enhancer, and IWR-1 and IWP-2 as inhibitors. 6-BIO is a molecule that amplifies the Wnt signal by specific pharmacological inhibition of glycogen synthase kinase 3 (GSK-3). By inhibiting GSK-3, β-catenin avoids degradation and induces morphogenic signals via the Tcf/Lef-1 cascade [16]. IWP-2 inhibits the activity of porcupine, a membrane-bound acyltransferase essential for Wnt protein production, and IWR-1 increases the expression of Axin protein, an inhibitor of Wnt/β-catenin pathway activity [17].

Further, it is well known that inflammatory cells play key roles during wound healing. Specifically, macrophages at the wound site, due to recruitment of blood-derived monocytes, are essential for effective wound healing. Previous studies have shown significantly impaired healing when macrophage infiltration is prevented [18]. In addition, fetal neutrophils are less likely to migrate than adult neutrophils, and the inflammatory response is reduced in fetal wound healing [19]. Therefore, in this study, the behavior of inflammatory cells in wound healing was investigated after treatment with Wnt inhibitors. Taken together, this study aimed to elucidate the relationship between wounds and Wnt/β-catenin signaling in mouse fetuses, to provide novel insights for the development of treatments related to scarless skin regeneration. The study results suggested that the Wnt signaling pathway may be related to wound healing without scar formation through stimulation of cell proliferation, dedifferentiation, and tissue remodeling.

## 2. Materials and Methods

The study protocol was reviewed and approved by the Keio University Institutional Animal Care and Use Committee of Keio University School of Medicine (approval number: 13072-(2)). All experiments were performed in accordance with Institutional Guidelines on Animal Experimentation at Keio University.

### 2.1. Fetal Wounding Procedure

Eight-week-old female ICR mice were used in this experiment. All mice were obtained from Sankyo Laboratory Services (Shizuoka, Japan). The mice were checked for vaginal plugs twice daily. When a plug was observed, the fetus was designated E0; the fetus was wounded at embryonic day 13 (E13), E15, and E17. Surgeries were performed on five pregnant mice per time point. Pregnant mice were anesthetized using 3% isoflurane, and the abdominal wall was incised to expose the uterus. Using an operating microscope, the myometrium and the amniotic and yolk sacs were incised. Subsequently, using surgical micro-scissors, a full-layer incision of approximately 2 mm in length was made in the lateral thoracic region of the fetus. On E13, after wounding, the amnion and yolk sac were sutured with 9-0 nylon, while the myometrium was left open and unsutured. The fetus was returned to the abdominal cavity with the amnion and yolk sac covered and the myometrium uncovered, and the abdomen was closed. Subsequently, just before the closure of both wounds, 1 μg/g body weight of the uterine relaxant ritodrine hydrochloride (Fujifilm Wako Pure Chemical, Osaka, Japan) was administered intra-peritoneally. The peritoneum and skin were then sutured with 5-0 nylon thread. The surgical methods used for each embryonic period were developed empirically [2]. The wound was labeled with 0.25% 1,1′-dioctadecy-3,3,3′,3′-tetramethylindocarbocyanine perchlorate (DiI) dissolved in 1% ethanol in phosphate-buffered saline (PBS) to mark the wound area and for visualization. Maternal mice were euthanized via cervical dislocation, and fetuses were harvested 24 h after wounding at each time point. Wounds were made in at least four fetuses. Fetal skin was harvested and fixed in 4% paraformaldehyde for 24 h and the fixed tissue was embedded in paraffin and stained. For frozen section immunostaining, the tissue was immersed in 20% sucrose/PBS after fixation, frozen, embedded in OCT compound (Sakura Finetek Japan Co., Ltd., Tokyo, Japan), and sliced at 7 µm thickness.

### 2.2. In-Situ hybridization

In-situ hybridization analysis was performed using the QuantiGene ViewRNA ISH Tissue Assay (Thermo Fisher Scientific, Waltham, MA, USA) according to the manufacturer’s protocol and as previously reported [20]. Briefly, the paraffin sections were dried at 60 °C for 60 min and paraffin was removed using Histo Clear (National Di-agnostics, Atlanta, GA, USA) and 100% ethanol. A hydrophobic wall was created around the tissues using ImmEdge Pen (Vector Laboratories, Burlingame, CA, USA). As pretreatment, the tissue was boiled in pretreatment solution for 5 min, rinsed, and treated with protease solution at 40 °C for 20 min. After washing twice with PBS, the samples were fixed in 10% neutral-buffered formalin solution for 5 min and washed again with PBS. The target probe was diluted 50-fold in probe-set diluent QF solution warmed to 40 °C and incubated at 40 °C for 3 h. After washing three times with wash buffer, the preamplifier solution was incubated at 40 °C for 25 min. The preamplifier solution was then washed three times with wash buffer again and incubated at 40 °C for 15 min. The AP enhancer solution was decanted and Fast Red Tablet was dissolved in naphthol buffer and incubated at room temperature for 5 min. After decanting the AP enhancer solution, Fast Red Tablet was dissolved in napthol buffer and incubated at 40 °C for 30 min. After washing twice with PBS, nuclear staining was performed with Gill’s Hematoxylin solution and washed with water three times. After removing excess staining solution with 0.01% ammonia solution, the samples were rinsed with water again and sealed in Ultramount Aqueous Permanent Mounting Medium (Agilent Technologies, Santa Clara, CA, USA).

### 2.3. Immunohistochemical Methods

#### 2.3.1. Paraffin Sections

In brief, the 7 µm thick paraffin sections were fixed with acetone for 3 min, and endogenous peroxidase was inactivated with 0.3% H_2_O_2_/MeOH for 30 min, followed by preparation of hydrophobic walls with ImmEdge Pen (Vector Laboratories). After washing with water, the sections were incubated for 10 min with a 0.2% Triton-X solution as an antigen activation treatment. After washing with PBS, blocking treatment was performed using the serum of secondary antibody-producing animals. Using the VECTASTAIN^®^ ABC KIT (Vector Laboratories), the primary antibody solution was diluted with PBS to an appropriate concentration. The following antibodies were used as primary antibodies: rabbit anti-Wnt1(1:100, Abcam, Cambridge, UK), rabbit anti-Wnt3 (1:200, Abcam), mouse anti-Wnt5a (1:200, Abcam), rabbit anti-Wnt5b (1:200, Abcam), rabbit anti-Wnt10b (1:200, Abcam), rabbit anti-Lef 1 (1:100, Abcam), rabbit anti-Axin2 (1:100, Abcam), and rabbit anti-β-catenin (1:250, Abcam). The sections were incubated at room temperature (15–25 °C) for 1 h and washed with PBS. Following this, color development with 3,3′-diaminobenzidine (DAB) solution was performed. The sections were then washed with water and nuclear staining with Gill’s Hematoxylin solution was carried out. Subsequently, the sections were washed with water, rinsed with 100% ethanol and 100% xylene, and then sealed using Mount-Quick (Takara Bio, Shiga, Japan). α-SMA staining was performed by autoimmunization with BOND MAX (Leica Microsystems K.K., Shinjuku, Tokyo, Japan) using the clone 1A4 antibody (1:1500, Agilent, Santa Clara, CA, USA).

#### 2.3.2. Frozen Sections

ICR mouse frozen skin sections (7 µm) were fixed with 100% acetone for 3 min and blocked with 5% BSA in PBS for 1 h at room temperature. Subsequently, the sections were stained with rabbit anti-Axin2 antibody (1:100, Abcam), mouse anti-F4/80 antibody (1:200, Bio-Rad, Richmond, CA, USA), and rat anti-Ly6g antibody (1:200, Abcam) overnight at 4 °C. Following this, Hoechst staining (1:500, Dojindo Lab, Kumamoto, Japan) was performed and incubated for 10 min at room temperature. Finally, the section was sealed, and the cells were counted using the Image J software.

### 2.4. Laser Micro Dissection (LMD), RNA Isolation, and Reverse Transcription

LMD was performed using a PALM MicroBeam (Carl Zeiss, Oberkochen, Germany). The manufacturer’s recommended slides and collection tubes (AdhesiveCap 500 opaque, Carl Zeiss) were set up, and the tissue was carefully cut after adjusting the aperture and intensity using a 20× magnification objective lens. The tube caps were filled with Buffer RLT (RNeasy Micro Kit; Qiagen, Hilden, Germany) and β-mercaptoethanol to allow separation of intact RNA. Total RNA was extracted from cells or skin tissues using a monophasic solution of phenol and guanidine isothiocyanate (ISOGEN; NipponGene, Tokyo, Japan) according to the manufacturer’s instructions. Total RNA was mixed with a random primer, reverse transcriptase, and dNTP mixture (Takara Bio, Tokyo, Japan). The mixture was then incubated in a T100TM thermal cycler (Bio-Rad Laboratories, Hercules, CA, USA) at 25 °C for 5 min, 55 °C for 10 min, and 80 °C for 10 min to heat inactivate the reverse transcriptase and synthesize cDNA.

### 2.5. Quantitative Real-Time Polymerase Chain Reaction (RT-PCR)

Quantitative real-time PCR was performed using an Applied Biosystems 7500 Fast Real-Time PCR System (Thermo Fisher Scientific). In total, 40 cycles were performed, and the fluorescence of each sample was measured at the end of each cycle. The PCR reaction was performed in two major steps: holding the reagent at 95 °C for 10 s (denaturation) and at 60 °C for 30 s (annealing and extension). In the subsequent melting curve analysis phase, the temperature was increased from 60 °C to 95 °C and fluorescence was measured continuously. Primers for Wnt1 (Mm01300555_g1), Wnt3 (Mm00437336_m1), Wnt5a (Mm00437347_m1), Wnt5b (Mm01183986_m1), Wnt10b (Mm00442104_m1), Axin2 (Mm01265780_m1), and Lef1 (Mm00550265_m1) were used (all Thermo Fisher Scientific). PCR master mix (Cat. 4352042; Applied Biosystems, Foster City, CA, USA) was used according to the manufacturer’s instructions. Actb (Mm02619580_g1) was used as the control gene for normalization according to the manufacturer’s instructions. Gene expression levels at normal sites were used as the baseline, and the fold-change values were determined using the 2−ΔΔCt method.

### 2.6. Wnt-Related Molecule Administration

6BIO ((2′Z,3′E)-6-bromoindirubin-3′-oxime, R&D Systems, Minneapolis, MN, USA) as a Wnt enhancer, IWR-1 (Sigma-Aldrich, St. Louis, MO, USA) and IWP-2 (Sigma-Aldrich) as Wnt inhibitors, each dissolved in dimethyl sulfoxide (DMSO) at a concentration of 10 mM, were administered topically in the amniotic fluid at the time of wounding of the E15 fetuses at 10 µL (10 µM) per fetus. Subsequently, 10 µL of DMSO was administered as the control. Then, 48 h later, the fetuses were collected and observed under a stereomicroscope (SZX16, Olympus Co., Ltd., Tokyo, Japan). In addition, Masson trichrome (MT) and Elastica van Gieson (EVG) staining were performed as previously reported [2].

### 2.7. Manchester Scar Scale (MSS)

The clinical, photographic, and histological evaluation methods of the MSS have been published [21]. These three methods have been suggested to be correlated, enabling effective quantification of the severity of a wide range of scars. The clinical evaluation method uses the Visual Analogue Scale, and the color, presence/absence of gloss, contour, distortion, and texture are evaluated and scored between 5 and 28 points (Appendix A), with a lower score indicating clinical improvement. In the histological method, the epidermis and dermis are divided, and the dermis is further divided into a papillary and reticular dermis, and evaluated using MT staining (Appendix A). Histological assessments are generally scored between 0 and 32 points, with lower histological scores indicating better results. However, the following considerations were applied to the histological evaluation method to score wound formation in this study: (1) due to the absence of keloid tissue in the mouse, score 5 was not used; (2) since the dermal papillae layer and reticular layer could not be distinguished from each other, evaluation was made in the dermis without distinction; (3) as the epidermis of all the wounds remained unchanged, epidermal scores were converted to 0. Based on these three changes, the scoring scale was adjusted to 0–12.

### 2.8. Statistical Analysis

The Mann–Whitney U test was performed to determine the significance of differences in migration or gene expression using the Statistica software version 9.0 (StatSoft, Tulsa, OK, USA). The results of descriptive statistics are presented as the mean ± standard deviation. The threshold for statistical significance was set at *p* < 0.05. Each experiment was performed in triplicate.

## 3. Results

### 3.1. Expression Levels of Wnt Molecules Vary at Different Developmental Stages

The skin is composed of multiple cell types, such as epithelial cells, fibroblasts, endothelial cells, and hair follicular cells. Therefore, we performed in-situ hybridization and RT-PCR on mouse fetal skin at various developmental stages to localize and quantify Wnt expression. Mouse fetuses were wounded at E13, E15, and E17; the wounds were collected 24 h later.

Wnt3 was specifically expressed in the epidermal layer (Figure 1). Further, RT-PCR analysis demonstrated significantly increased expression at the wound site compared to the normal site at E13 (*p* = 0.015) and E15 (*p* = 0.026), which decreased at E17 (*p* = 0.025). Wnt5a expression throughout the dermal layer was confirmed by immunostaining and expression was also found in the hair follicles (Figure 2). However, RT-PCR results showed decreased expression at E13 in the wounds when the skin was completely regenerated (*p* = 0.0096).

The expression of Wnt5b in the dermal layer was also confirmed (Figure 3). The expression was enhanced at E13, E15, and E17 in the normal tissues as compared to the wound. Further, RT-PCR confirmed decreased expression at the wound site compared to the normal site at E15 (*p* = 0.0073) and E17 (*p* = 0.0061). However, there was no difference observed at E13, suggesting that Wnt5b expression may be related to skin regeneration.

In this study, Wnt1 expression was not observed from E13 to E17 at the mRNA level (Appendix A). However, Wnt10b expression was observed in the epidermal basal layer around the wound at E13, and on the entire dermal layer at E15 and E17 (Appendix A).

At E13, nuclear localization of β-catenin at the wound sites was not different from that in the normal sites (Figure 4). In addition, at E15 and E17, partial nuclear accumulation was observed in the basal layer of the normal tissue, but not at the wound site. In-situ hybridization of Axin2 (Figure 5) demonstrated specific expression around the basement layer at E13, which was slightly decreased at the wound site. Additionally, significantly decreased Axin2 expression was found at the wound site at E15 (*p* = 0.015) and E17 (*p* = 0.0043), which was also confirmed by RT-PCR.

Lef1 expression was specifically expressed around the basement layer at E13, which was slightly decreased at the wound site (Figure 6). At E15, it was specifically expressed in the basal layer and hair follicle, with significantly decreased expression at the wound site. At E17, in the normal tissues, expression was lower, but specific expression in the hair follicle was observed. However, expression was similarly decreased at the wound site. Moreover, RT-PCR analysis at E15 (*p* = 0.0049) and E17 (*p* = 0.00061), compared to the normal tissue, demonstrated a clear decrease in Lef1 expression level at the wound site.

### 3.2. Improved Wound Healing upon Wnt Enhancer/Inhibitor Administration

To evaluate whether Wnt signaling influenced morphological changes during wound healing, Wnt modulators were administered to E15 fetal mice in the amniotic fluid. Administration of IWP-2 regenerated the skin structure and healed without leaving visible marks; administration of IWR-1 reduced visible marks in the scar area compared to the control; administration of 6-BIO showed no changes (Figure 7A). When the wound tissue structure was observed following hematoxylin staining, regeneration of the hair follicle and the structural arrangement of the skin were impaired at the wound site. In the tissue treated with 6-BIO, the structure of the dermis was disturbed, regeneration of the hair follicle was not observed, and the thickness of the dermis was reduced. In contrast, in IWR-1- and IWP-2-treated tissues, hair follicle regeneration was confirmed, suggesting that there was less damage to the dermal structure compared to the control (Figure 7B). Further, EVG staining suggested that collagen fibers were only slightly damaged in the wound area of the IWR-1 treatment group (Figure 7C).

In the 6-BIO-treated groups, the expression of α-SMA, a marker that increases remarks at the wound during scar formation in the smooth muscles and blood vessels, was enhanced at the perimeter of the wound (Figure 7D,E), which was attenuated in the IWR-1 and IWP-2 administration groups. In addition, MT staining suggested that fibrosis was promoted at the wound edge of the 6 BIO administration group. In contrast, in the IWP-2 and IWR-1 administration groups, compared to normal tissues, fibrosis was suppressed. Further, administration of IWR-1 was shown to significantly reduce the number of α-SMA-positive cells (*p* = 0.00015) at the wound site and suppress tissue fibrosis.

In the MSS evaluation, both the IWR-1 (*p* = 0.0000059) and IWP-2 (*p* = 0.00029) groups showed significant decreases in scores compared to the control group (Figure 7F), indicating that scar formation was significantly suppressed with Wnt inhibitor treatment.

### 3.3. Inflammatory Cells May Mediate Wnt-Regulated Changes in Wound Healing

The inflammatory response is thought to be related to the difference in tissue structure; therefore, F4/80 (macrophage marker) and Ly-6g (neutrophil marker) were stained by a fluorescent antibody method (Figure 8). F4/80-positive cells were found in the wounds at both E15 and E17, and were enhanced in the IWR-1-treated group compared to the control group, and were significantly increased at E17 (*p* = 0.021). In other words, an increase in macrophages in the treated group was suggested. Conversely, Ly-6g expression in the IWR-1 treatment group was attenuated, suggesting that neutrophil expression was decreased by E17. From the cell count assay, it was found that F4/80-positive cells were significantly increased in the IWR-1 treatment group, while Ly-6g-positive cells were decreased, although there was no significant difference (*p* = 0.094).

## 4. Discussion

Wnt plays many roles in various aspects of skin wound repair. In addition to the formation of skin appendages, such as hair, it is involved in the construction of epithelial structures and the reconstruction of the dermal compartment. β-catenin is an important regulator of fibroblast behavior. Although Wnt signaling is not essential for maintaining elevated levels of β-catenin during the proliferative phase of skin healing, it has been reported to be involved in β-catenin stimulation [22].

In this study, we compared the expression pattern of Wnt-related molecules and fetal wound healing at E13, E15, and E17. E13 wound healing is an ideal model of regeneration during morphogenesis, including skin texture and hair follicles. E15 wound healing does not regenerate skin texture, but hair (partially) and alignment of the ECM are regenerated, thus, serving as a model of dermal regeneration. In contrast, E17 wounds do not regenerate and instead leave scars, which is a model of dermal fibrosis. By comparing these models of wound healing at different developmental stages, the mechanism of epidermal morphogenesis, dermal regeneration, and scarring can be analyzed.

In this study, we analyzed the expression of Wnt family and β-catenin-related molecules during wound healing in mice at each developmental stage. The Wnt/β-catenin pathway is mainly divided into canonical Wnt signaling (β-catenin/T cell factor (TCF)-dependent interaction) and non-canonical Wnt signaling (β-catenin/TCF-independent interaction). In mammals, 19 Wnt genes have been identified, and Wnts 1, 2, 3a, 6, 7a, 7b, 8a, 9a, 10a, and 16 mainly activate the canonical pathway, while Wnts 4, 5a, 5b, and 11 activate the non-canonical pathway [23].

Expression of Wnt3 was amplified at the wound site on E13 and E15, but the expression was similar to that in the normal skin at E17. The Wnt3 protein is thought to play an important role in determining the anterior–posterior axis early in embryonic development [24], but its role in wound healing is unknown. Increased expression in early fetal wounds suggests that the canonical Wnt pathway was activated in E13 and E15 wounds; thus, it may be related to epidermal and dermal regeneration. In other organs, Wnt3a represents the classical canonical Wnt ligand that triggers β-catenin signaling [25], but no studies have examined whether it exerts its effects via β-catenin in fibrosis or scarring. Therefore, further verification is required to confirm this result.

In E13, when skin regeneration occurred without scarring, Wnt5a expression was reduced compared to that in the normal skin, and nearly identical expression was observed at E15 and E17 during regulation of skin texture and dermal fibrosis. Wnt5b shares 80% amino acid identity with Wnt5a at E13, but the functional difference is not clear [26]. The Wnt5a protein is known to inhibit the β-catenin pathway, although it is also known to activate the non-canonical pathway. The decrease in Wnt5a expression at the wound site on E13 indicates that the canonical pathway was activated during E13 wound healing. The expression of Wnt5b, which was reduced at the wounded site on E15 and E17, may indicate that the non-canonical pathway was involved in scar formation. However, purified Wnt5a protein activates or inhibits the β-catenin TCF signal, depending on the state of the receptor [27]. Therefore, the type of Wnt signal may not determine the specificity of the pathway, which may explain the conflicting Wnt5a and Wnt5b results.

Furthermore, the expression of Axin2, a molecule involved in the Wnt/β-catenin pathway, and Lef1 showed expression patterns similar to those in the normal sites in the normal embryo wounds, but were attenuated in the wounds after E15. Axin is a key molecule in the Wnt signaling pathway that suppresses signaling activity in the absence of a Wnt ligand. Axin is thought to exert a negative feedback function on Wnt signaling by promoting β-catenin degradation in cooperation with APC and GSK3β [15,28]. Wnt lymphocyte enhancer binding factor 1 (LEF1) is a member of the high-mobility group transcription factor T cell factor (TCF)/LEF1 family, a downstream mediator of the pathway [29,30,31]. As Axin and LFF1 cannot be functionally evaluated by their expression, the expression of these molecules was altered around wounds, demonstrating the impact of the Wnt signaling pathway before and after the transition.

At the wound site of E13 mice, β-catenin was not expressed and translocated to the nucleus as in the normal skin but was translocated to the nucleus in the basal layer of the normal skin in the late embryo. In the canonical pathway, β-catenin translocates to the nucleus when a Wnt ligand binds to its receptor to perform various functions. At E13, the canonical pathway is usually not activated in the wound site, as in the unwounded site, suggesting that the tissue does not change its status before and after wound healing. Thus, complete regeneration may occur at E13.

Scar formation after wound healing has been shown to be improved by Wnt inhibitor (IWR-1 and IWP-2) administration. IWP was first discovered as an inhibitor of Wnt production [18]. For a Wnt protein to be functionally active, it must undergo palmitoylation during biosynthesis [15]. This acylation process adds monosaturated fatty acid (Z) -hexadec-9-enoic acid to the highly conserved serine residue 209. Inhibition of Wnt signaling requires prevention of ligand secretion, which is required for palmitoylation of Wnt ligands at the highly conserved serine residue 209 before secretion of the endoplasmic reticulum-resident enzyme porcupine (PORCN). IWR-1 and IWP-2 have the same core structure as a benzothiazol-2-amine moiety, which is important for inhibiting PORCN function, resulting in the suppression of Wnt signaling [32]. In-vivo studies have shown that IWP compounds block the Lrp6 receptor, Dvl2 phosphorylation, and β-catenin accumulation, and IWR compounds promote β-catenin degradation by stabilizing Axin2, indirectly targeting regulatory events downstream of Lrp6 and Dvl2 [18]. It has been suggested that the Wnt/β-catenin pathway is involved in skin inflammation and fibrosis. In vivo, transgenic mice overexpressing Wnt-10b in mesenchymal cells have been shown to enhance the nuclear translocation of β-catenin and develop spontaneous dermal fibrosis [33]. In vitro, adenovirus-mediated overexpression of Wnt-3a promotes the profibrotic phenotype of fibroblasts and increases proliferation, migration, and differentiation [34]. This suggests that some inhibition of the Wnt family is involved in fibrosis, and that inhibition of Wnt signaling suppresses inflammation and fibrosis, suggesting the possibility of scarless wound healing.

In addition, the suppression of the Wnt-β catenin signaling pathway by IWR-1 administration resulted in the suppression of fibrosis in the wound in the late embryo, resulting in decreased monocyte accumulation and increased macrophage accumulation. Macrophages are involved in the innate immune response and play a variety of roles in the inflammatory process [35]. Their accumulation in wounds was suppressed in this study, demonstrating that some subtypes of macrophages (M2 macrophages) may suppress fibrosis. In vitro, the polarization of human macrophages to the M2 phenotype has been shown to be associated with increased levels of Wnt1 and Wnt3a in macrophages [36]. It suggested that some Wnt signals activate macrophages to suppress fibrosis in wounds. Further elucidation of the macrophage subtypes is needed to understand this mechanism.

A limitation of our study is that we focused only on Wnt signaling during fetal wound healing. Our previous observations point out that wound healing in the mouse fetus is epithelialized at 48 h [2], but the behavior of Wnt at an earlier stage, or after scar formation, and the effects of inhibitors need to be examined in the future.

In this study, we characterized Wnt expression at wound sites in a mouse wound-healing model and observed the wound-healing process using a Wnt inhibitor. These data suggested that the Wnt signaling pathway may be related to wound healing without scar formation through stimulation of cell proliferation, dedifferentiation, and tissue remodeling. Further research on the role of Wnt signaling is needed to develop a new strategy for scarless wound healing.

## Figures and Tables

**Figure 1 biomedicines-10-01536-f001:**
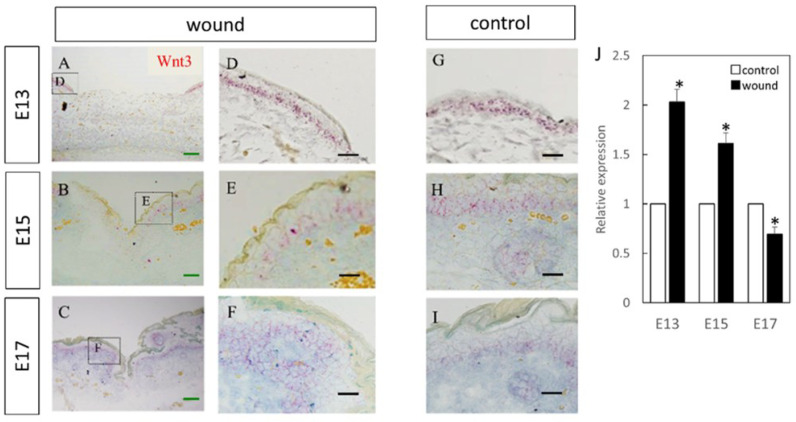
Measurement of Wnt3 expression in the wound by in-situ hybridization and real-time quantitative PCR. Wnt3 expression is indicated in red in the tissue. Wnt3 was specifically expressed in the epidermal layer. (**A**–**C**) Weakly magnified images of the wound perimeter. (**D**–**F**) Strongly magnified images of the wound margin. (**G**–**I**) Observed images of the normal area. (**J**) Decreased expression of Wnt3 at the wound site compared to the normal site at E17. In contrast, Wnt3 expression was increased at the wound sites on E15 and E13. * *p* < 0.05. (**A**–**C**) Scale bar = 200 μm. (**D**–**I**) Scale bar = 100 μm.

**Figure 2 biomedicines-10-01536-f002:**
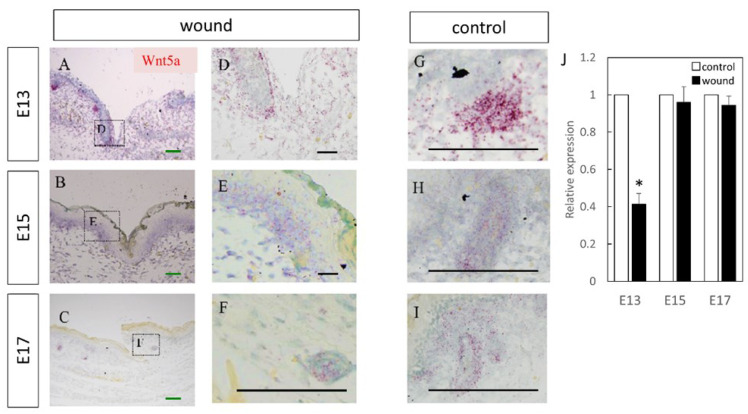
Measurement of Wnt5a expression in wound by in-situ hybridization and real-time quantitative PCR. Wnt5a expression is indicated in red in the tissue. Wnt5a was specifically expressed in the dermal layer around the epidermal-dermal junction and around hair follicles. (**A**–**C**) Weakly magnified images of the region around the wound. (**D**–**F**) Strongly magnified images of the wound margin. (**G**–**I**) Observed images of the normal area. (**J**) Decreased expression of Wnt5a at the wound site on E13. * *p* < 0.05. (**A**–**C**) Scale bar = 200 μm. (**D**–**I**) Scale bar = 100 μm.

**Figure 3 biomedicines-10-01536-f003:**
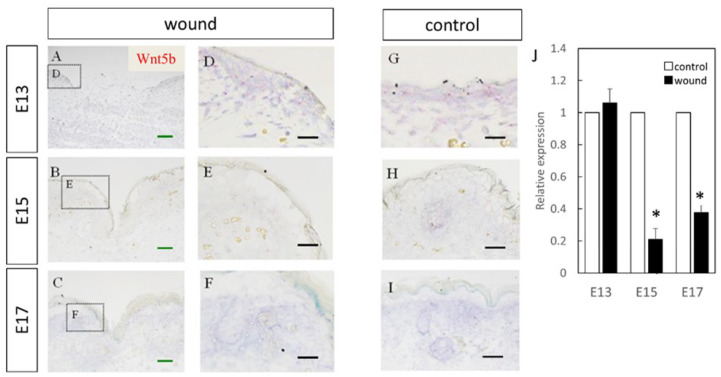
Measurement of Wnt5b expression in the wound by in-situ hybridization and real-time quantitative PCR. Wnt5a expression is indicated in red in the tissue. Wnt5b expression was confirmed in the dermal layer. Expression was increased in E13, but in E15 and E17, Wnt5b expression in the wound area was decreased compared to that in the normal areas. (**A**–**C**) Weakly magnified images of the region around the wound. (**D**–**F**) Strongly magnified images of the wound margin. (**G**–**I**) Observed images of the normal area. (**J**) Expression at the wound site was decreased on E15 and E17. * *p* < 0.05. (**A**–**C**) Scale bar = 200 μm. (**D**–**I**) Scale bar = 100 μm.

**Figure 4 biomedicines-10-01536-f004:**
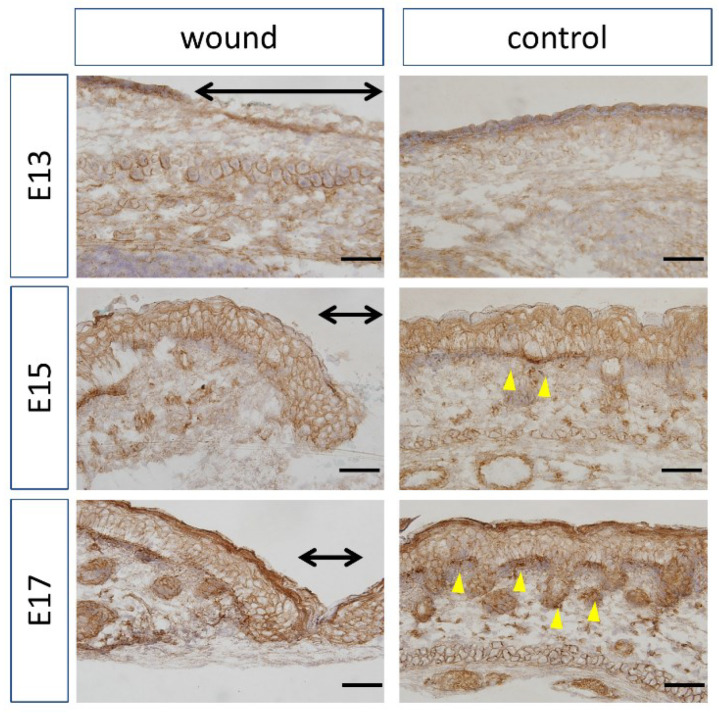
Localization of β-catenin by immunohistochemistry. The range of the wound is indicated by a yellow arrow. Brown: β-catenin. Purple: Hematoxylin. Black arrow: Area of the wound. Scale bar = 100 μm.

**Figure 5 biomedicines-10-01536-f005:**
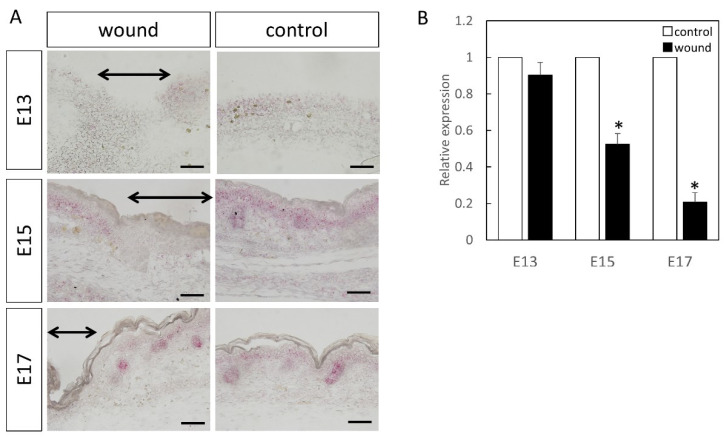
Localization of Axin2 (Red) in the wound. The range of the wound is indicated by an arrow. Results of Axin2 localization by (**A**) in-situ hybridization and (**B**) real-time PCR. At E13, expression did not change at the wound site, but decreased on E15 and 17. * *p* < 0.05. black arrow; area of the wound. Scale bar = 100 μm.

**Figure 6 biomedicines-10-01536-f006:**
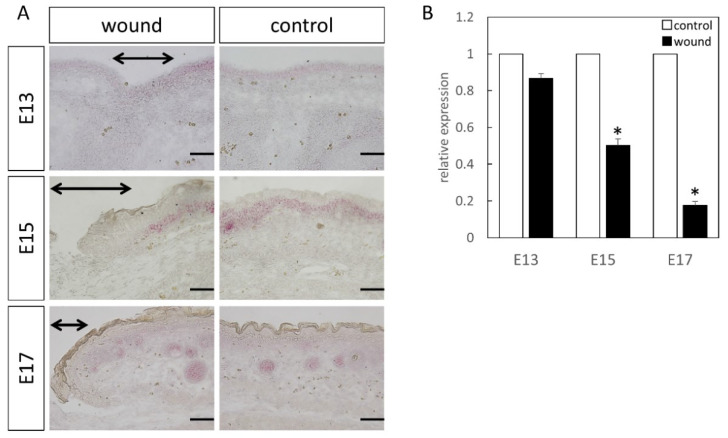
Localization of Lef1 (Red) in the wound. The range of the wound is indicated by an arrow. Results of Lef1 localization by (**A**) in-situ hybridization and (**B**) real-time PCR. Expression did not change at E13. However, on E15 and E17, significantly decreased expression was observed at the wound site. * *p* < 0.05. Scale bar = 100 μm.

**Figure 7 biomedicines-10-01536-f007:**
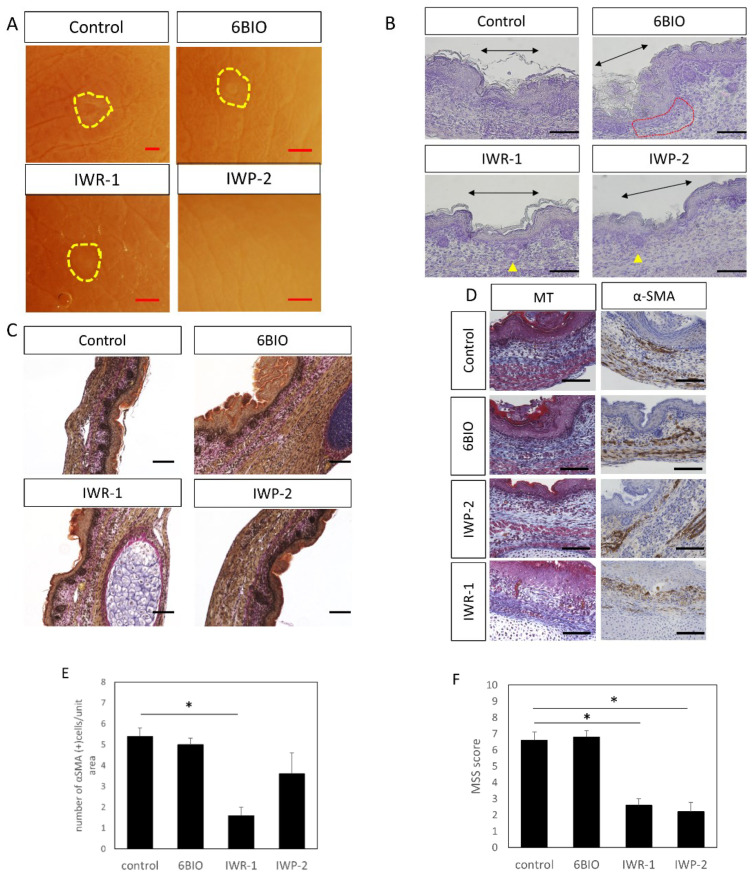
E15 scar healing by administration of Wnt enhancer/inhibitor. (**A**) Observation by stereomicroscope. The scar is indicated by a yellow circle. Scars were not noticeable in the IWP-2 administration group. Scale bar = 1 mm. (**B**) Hematoxylin staining, scale bar = 100 μm. Black arrow: area of the scar. Yellow triangle: follicular structures observed within the scar. Red dotted line: area of thinning of the dermal structure in the scar. (**C**) EVG staining, scale bar = 100 μm. (**D**) Masson-Trichrome and α-SMA staining, scale bar = 100 μm. (**E**) Cell counting of α-SMA positive cells in wound. (**F**) Relationship between Wnt enhancer/inhibitor administration and wound MSS. * *p* < 0.05.

**Figure 8 biomedicines-10-01536-f008:**
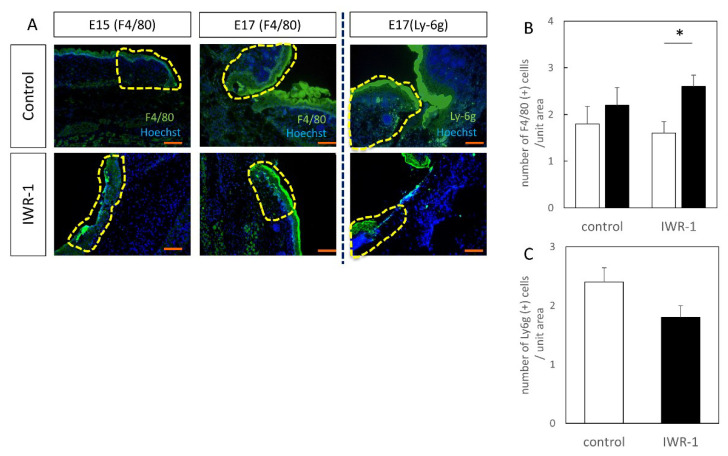
Localization of inflammatory cells in wound. (**A**) Immunofluorescence staining, scale bar = 100 μm. (**B**) F4/80-positive cell counting. F4/80-positive cells were significantly increased in the E17-48H IWR-1 treatment group. (**C**) Ly-6g-positive cell counting. * *p* < 0.05.

## Data Availability

Data is contained within the article and Appendix A.

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
