# Peer review of "Role of Wnt Signaling in Mouse Fetal Skin Wound Healing"

_biomedicines, 2022, doi:10.3390/biomedicines10071536_

Round 1

Reviewer 1 Report

The authors should must indicate better the acronyms.

In the results paragraph  3.1, the authors should have to describe better the results pointing the difference of tissue and consequently the dye differences. For example it is not clear what is the difference in figure 1 of a,b,c....j. 

page 9:improved wound healin upon Wnt .....line 392 explain bettere why the thickness is reduced  also adding arrows or symbols. It is very difficult to follow the figures. in the figure 7a it is not evident the improved skin texture because the images seemed to be in focus. The discussion is well written 

Author Response

Response to reviewer #1:

The authors should must indicate better the acronyms.

-As you pointed out, the correct expansion of WIHN is wound-induced hair follicle neogenesis and we have corrected it (page 2, line 57.

In the results paragraph  3.1, the authors should have to describe better the results pointing the difference of tissue and consequently the dye differences. For example it is not clear what is the difference in figure 1 of a,b,c....j.

- As you pointed out, to the legends of Figures 1, 2, and 3, we have added the differences in pigmentation and organization (page.6, line 258-260, 265-268; page.7, line 277-281).

page 9:improved wound healing upon Wnt .....line 392 explain better why the thickness is reduced  also adding arrows or symbols. It is very difficult to follow the figures. in the figure 7a it is not evident the improved skin texture because the images seemed to be in focus. The discussion is well written.

- Thank you for your comments. We have added symbols and lines to Figure 7c showing the site of follicular formation within the scar and the thinning of the dermis in the scar area due to 6-BIO administration. In addition, in the description of Figure 7a, we have included the structure of the skin texture and the degree of visible marks and have revised the description to make the level of healing  easier to understand (page.9, line 317-319).

Reviewer 2 Report

In my opinion, this study presented in manuscript "Role of Wnt signaling in mouse fetal skin wound healing " is well designed. 

Interesting and not trivial in execution experiments with the application of wounds at critical times of embryonic development, assessment of changes in the key participants of Wnt/beta-catenin signaling pathway were carried out.

Finally, the authors tested the possibility of modulating the observed processes using 2 inhibitors and one activator of the Wnt-signaling pathway. The clear statement of the would healing problem, the description of the key signaling participants, as well as the aim of authors in the Introduction section, it is easy to further navigate the results and interpretation gived by authors. 

The description of the results of the study is complemented by high quality figures with two method for estimation expression of key members Wnt-pathway.

The authors quite rightly state in the final part of the article that the developed model has a number of advantages compared to traditional wounding models.

I consider it reasonable to accept such a manuscript for publication. 

Author Response

Response to reviewer #2:

In my opinion, this study presented in manuscript "Role of Wnt signaling in mouse fetal skin wound healing " is well designed.

Interesting and not trivial in execution experiments with the application of wounds at critical times of embryonic development, assessment of changes in the key participants of Wnt/beta-catenin signaling pathway were carried out.

Finally, the authors tested the possibility of modulating the observed processes using 2 inhibitors and one activator of the Wnt-signaling pathway. The clear statement of the would healing problem, the description of the key signaling participants, as well as the aim of authors in the Introduction section, it is easy to further navigate the results and interpretation gived by authors.

The description of the results of the study is complemented by high quality figures with two method for estimation expression of key members Wnt-pathway.

The authors quite rightly state in the final part of the article that the developed model has a number of advantages compared to traditional wounding models.

I consider it reasonable to accept such a manuscript for publication.

-We appreciate your thoughtful review and comments.